# Capturing Static, Short-Term, and Long-Term Dynamics Through Self-Supervised Time Series Learning: CHRONOS

## Abstract

Time series data presents a rich tapestry of temporal patterns, encompassing enduring static trends that persist throughout the temporal sequence and dynamic patterns that define its evolving nature. Adopting a comprehensive approach that considers these distinct temporal facets is essential to advance the field of Self-Supervised Learning (SSL) in time series analysis. In this paper, we introduce the Contrasting Heads Represent Opposed Natures of Signals (CHRONOS), a novel SSL methodology that drives the model to understand three distinct temporal attributes – static, short-term, and long-term dynamics. This is achieved by projecting the representations into two separate spaces, employing contrasting heads. Furthermore, selective optimization leads distinct model units to be specialized in different temporal natures. CHRONOS is evaluated by applying this methodology to the analysis of electrocardiogram (ECG) signals across four distinct downstream tasks, utilizing four independent datasets. The study demonstrates the consistent performance of CHRONOS across all tasks, surpassing the state-of-the-art methods for time series analysis. CHRONOS serves as a testament to the importance of capturing diverse temporal aspects of time series data for driving versatile models capable of consistently excelling in a wide spectrum of downstream tasks.

## 1 Introduction

The application of signal processing to physiological signals opens up a wide range of opportunities for health-related analysis, including but not limited to arrhythmia detection (Ebrahimi et al., 2020), gender identification, and age estimation (Attia et al., 2019b). It is reasonable to consider that these distinct analysis tasks necessitate the extraction of distinct signal trends. More precisely, tasks such as gender identification would require to focus on capturing the static patterns present within the signals, while for arrhythmia detection, the emphasis should be on identifying and analyzing the specific dynamic patterns associated with irregular cardiac rhythms. As such, *different tasks require capturing different temporal aspects of the time series data*. We identify three distinct temporal aspects: the static, the short-term dynamics and the long-term dynamics.

Different Self-Supervised Learning (SSL) methods tailored for time series data have been recently published such as (Wickstrøm et al., 2022), Diamant et al. (2022), or (Zhang et al., 2022). However, considering these three different temporal natures together as part of their training objective has not been explored in the field. We hypothesize that incorporating the different temporal attributes inherent in the signal will drive a SSL methodology to encode valuable patterns for addressing a wide range of diverse downstream tasks. For this purpose, we present Contrasting Heads Represent Opposed Natures of Signals (CHRONOS), which is a novel SSL method that is designed for understanding patterns belonging to these three distinct temporal dynamics; static, short-term, and long-term. This is achieved by: (i) Projecting the representations into two different spaces using contrasting heads, (ii) Defining different loss functions for each of them, which drives the model to represent a particular temporal nature, and (iii) Carrying out a selective optimization, in which the most important part of the model, responsible for the temporal nature, is considered.

To assess the performance of CHRONOS, we carried out four different experiments on four different databases. For each experiment, a Machine Learning (ML) model is fitted on top of the representations computed by the pre-trained encoder. The performance of CHRONOS is compared against four state-of-the-art methods in the field. We demonstrate that CHRONOS achieves excellent performance in two out of the four experiments, and notably, it consistently attains good results across all four distinct experiments, marking it as the sole approach to showcase this generalization proficiency. In summary, the contributions of this paper are:

1. We introduce CHRONOS, a novel SSL method that considers different temporal dynamics that enforce the representation to contain static as well as short-term and long-term dynamic patterns presented in the time series data.
2. We show that by understanding these different temporal aspects, the model is able to perform consistently in four different downstream tasks, evaluated on four different datasets.
3. We open a new avenue in which the different natures of the time series data are considered for driving the model to be generic and performant across different downstream tasks.

## 2 RELATED WORK

Interpreting physiological signals demands a level of expertise, making the labeling of such data costly. It represents a bottleneck within the deep learning context, particularly given the substantial volume of data required for training deep learning models. Consequently, various studies have been directed towards optimizing the model using a few amount of labels (Fan et al., 2021) or directly without relying on these annotations (Tonekaboni et al., 2021) and (Franceschi et al., 2020).

With the rise of SSL, different methods have recently been designed in the field of time series analysis, exploiting the characteristics of the time series data. Contrastive Learning of Cardiac Signals Across Space (CLOCS) (Kiyasseh et al., 2021) exploits the common information between different ECG leads. The Mixing-up method (Wickstrøm et al., 2022) utilizes the temporal characteristics of the data for a more tailored data augmentation by creating a time series, which is a product of mixing two time series from the same subject. An alternative approach is proposed in the Time-Frequency Consistency (TF-C) (Zhang et al., 2022) method, which produces two variations of the time-domain and frequency-domain pairs associated with the input signal. SSL method is developed to identify similarities inherent in these paired representations.

In a more promising manner, the Patient Contrastive Learning (PCLR) method (Diamant et al., 2022) obviates the need for data augmentation by leveraging the inherent dynamic nature of physiological signals. It avoids using data augmentation by considering two time series belonging to the same subject as positive pairs. Distilled Encoding Beyond Similarities (DEBS) (Atienza et al., 2023) continues this line of research. By using a non-contrastive method, it incorporates a dissimilarity metric to capture the short-term dynamic features contained in the time series data. CHRONOS extends both of these previous studies by capturing long-term dynamics as a novel temporal dimension which plays a crucial role in the performance of the representation for all the downstream tasks.

## 3 CONTRASTING HEADS REPRESENT OPPOSED NATURES OF SIGNALS (CHRONOS)

The underlying concept of CHRONOS is the identification of three distinct temporal characteristics within time-series data; static, and short-term and long-term dynamics:

**Static** – This pertains to characteristics in a time-series signal that do not change over time. For example, since heart rate can be used as a biometric identifier Ramli et al. (2016)), we can assume that there are consistent patterns in heart rate signals that do not change over time.

**Short-Term Dynamics** – This pertains to the more volatile nature of the signals, characterized by patterns that undergo gradual changes over time. For example, in ECG signals these are the variations due to the beating of the heart and show variation over seconds and minutes.

**Long-Term Dynamics** – These are patterns that are not static but evolve slowly over time, like days, months, or years. In the biomedical field, these patterns may be related to age, body weight, cholesterol level, or other characteristics that change over months or years.

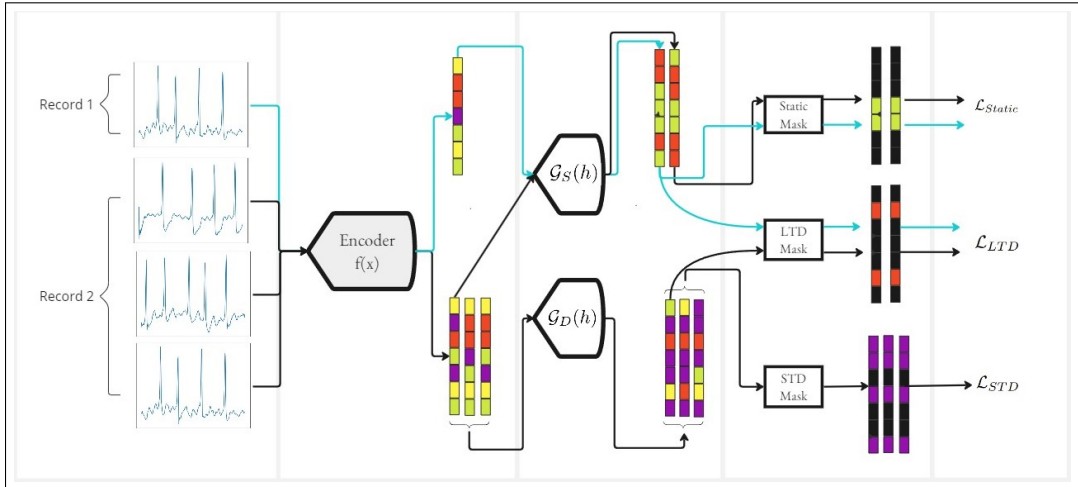

Figure 1: CHRONOS illustrated. Different colors represent different temporal natures: Green (static), purple (short-term dynamics), and red (long-term dynamics). The different colors of the arrows indicate inputs from different recordings. While $\mathcal{L}_{Static}$ and $\mathcal{L}_{LTD}$ are computed between the time series representations belonging to different records, $\mathcal{L}_{STD}$ is computed using time series belonging to the same record. The four inputs belong to the same subject.

CHRONOS is designed to capture these three distinct temporal dimensions in a unique representation and is illustrated in Figure 1. The model takes up to four different time strips belonging to two different records from the same subject. The encoder computes the representations of these distinct inputs. The Static Projector (labeled $\mathcal{G}_S$) takes a pair of representations from the two different recordings which are displayed in a Static Space. The static loss function ($\mathcal{L}_{Static}$) is computed between these pair of projections, ensuring that the unique patterns of each subject are represented consistently even between different records. The Short-term Dynamic Projector (labeled $\mathcal{G}_D$) displays the three representations from the same recording in the Short-Term Dynamic Space. These Short-term Dynamic Projections are used for computing the loss function for short-time dynamics ($\mathcal{L}_{SDT}$). In addition, the method incorporates the long-term dynamic loss function ($\mathcal{L}_{LTD}$), which is computed between a pair of projections from the two distinct spaces (further detailed in Section 3.2.1). This loss function encourages the model to represent the long-term dynamics between time series separated by a significant temporal gap. Finally, the conflicting nature of $\mathcal{L}_{Static}$ and $\mathcal{L}_{LTD}$ is alleviated by a selective optimization (described in Section 3.2.2). The $Static\ Mask$, $LTD\ Mask$, and $STD\ Mask$ shown in Figure 1 ensure that only the most meaningful components of each projection are used for computing the three loss functions. CHRONOS inherits the use of the two projectors and the $\mathcal{L}_{Static}$ and $\mathcal{L}_{SDT}$ functions from DEBS[1]. The use of, and linkage to DEBS is presented first in Section 3.1.

## 3.1 Revisiting Distilled Encoding Beyond Similarities (DEBS)

CHRONOS is built upon DEBS (Atienza et al., 2023). For a better understanding of this manuscript, we believe it is indispensable to briefly review the commonalities of DEBS with this study. However, this is not a direct contribution of this manuscript.

### 3.1.1 Non-Contrastive Method

It is crucial to emphasize that all four inputs used for the optimization originate from the same subject. Therefore all of them are positively linked. This is the reason that we consider them as positive pairs, notwithstanding the optimization of at least one cost function to accentuate distinctions between two of the four inputs. In this context, the non-contrastive framework from BYOL (Grill et al., 2020) is employed, featuring both a teacher network and a student network, each with

---

[1]To align with the context of this paper, we opt to rename the original DEBS loss functions "Similarities Loss" and "Gradual Loss" as "Static Loss" and "Short-Term Dynamics Loss," respectively.

an encoder and a projector initialized with identical parameters. The student network is additionally equipped with a predictor, which acts on the views computed by the projector. The optimization involves Stochastic Gradient Descent (SGD) for the student network, projector and predictor. Both the teacher network and the teacher projector serve as an exponential moving average (EMA) of the student network.

It is worth mentioning that both the projector and the predictor are discarded after the optimization process. Therefore, the representations used for the downstream tasks are obtained directly from the student network. The student-teacher framework and the characteristic predictor are excluded from Figure 1 for clarity. In practise, the different loss functions are computed between projection/prediction pairs obtained by the teacher and student networks, denoted as $\mathbf{z}$ and $\mathbf{q}(\zeta)$, where $\zeta$ represents the student projection.

### 3.1.2 PROJECTING REPRESENTATIONS INTO DIFFERENT SPACES

To enhance model learnability, CHRONOS projects representations into two distinct spaces, similar to the Multi-Head mechanism in Transformer models (Vaswani et al., 2017). This method computes different loss functions in these spaces, represented as $\mathcal{G}_s$ and $\mathcal{G}_d$ in Figure 1. The two spaces are detailed below:

**Static Space:**  This space emphasizes the persistence of specific patterns throughout the entire recording in representations' projections. The "Static Loss" is defined as:

$$\mathcal{L}_{Static}(\mathbf{z}_2^s, \mathbf{q}(\zeta_1)^s) = 1 - \frac{\mathbf{z}_2^s \cdot \mathbf{q}(\zeta_1)^s}{\max\left(\|\mathbf{z}_2^s\|_2 \cdot \|\mathbf{q}(\zeta_1)^s\|_2, \epsilon\right)}, \tag{1}$$

where $\mathbf{z}_2^s$ and $\mathbf{q}(\zeta_1)^s$ are the static projection and static prediction for inputs belonging to the first and second records.

**Short-Term Dynamic Space:**  In contrast to the static space, this space directs attention towards projections of volatile patterns throughout the recording. The "Short Term Dynamic Loss" ($\mathcal{L}_{STD}$) ensures smooth evolution of representations over time, capturing changes in temporal data. It is defined as:

$$\mathcal{L}_{STD}(\mathbf{q}(\zeta_t^d), \mathcal{PAR}(\mathbf{z}_{t-i}^d, \mathbf{z}_{t+j}^d)) = 1 - \frac{\mathbf{q}(\zeta_t^d) \cdot \mathcal{PAR}(\mathbf{z}_{t-i}^d, \mathbf{z}_{t+j}^d)}{\max\left(\left\|\mathbf{q}(\zeta_t^d)\right\|_2 \cdot \left\|\mathcal{PAR}(\mathbf{z}_{t-i}^d, \mathbf{z}_{t+j}^d)\right\|_2, \epsilon\right)}, \tag{2}$$

where $\mathbf{z}_{t-i}^d$, $\mathbf{q}(\zeta_t^d)$, and $\mathbf{z}_{t+j}^d$ are the dynamic projections/predictions of the three consecutive inputs drawn from the same record. Pondered Average Representation (PAR) is the approximation of $\mathbf{z}_t$, drawn from $\mathbf{z}_{t-i}$ and $\mathbf{z}_{t+j}$. It is calculated as,

$$\mathcal{PAR}(\mathbf{z}_{t-i}^d, \mathbf{z}_{t+j}^d) = \frac{\mathbf{z}_{t-i}^d \cdot j + \mathbf{z}_{t+j}^d \cdot i}{i + j}, \tag{3}$$

where i and j represent the time delay between the different inputs.

### 3.2 THE METHOD

### 3.2.1 LONG TERM COMPONENT

To effectively model the long-term dynamics, the CHRONOS framework introduces a novel loss function, named the "Long Term Loss ($\mathcal{L}_{LTD}$)". This function operates by examining the temporal data segments pertaining to the same individual, which are separated by a sufficient time interval for long-term changes to occur. The primary objective of $\mathcal{L}_{LTD}$ (Eq.(4)) is to encourage dissimilarity between pairs of representations, thereby capturing alterations between long-term dynamic patterns over time.

$$\mathcal{L}_{LTD}(\mathbf{z}_1^s, \mathbf{q}(\zeta_2)^d) = 1 + \frac{\mathbf{z}_1^s \cdot \mathbf{q}(\zeta_2)^d}{\max\left(\|\mathbf{z}_1^s\|_2 \cdot \|\mathbf{q}(\zeta_2)^d\|_2, \epsilon\right)}, \tag{4}$$

where $\mathbf{z}_1^s$ and $\mathbf{q}(\zeta_2)^d$ are the static projection and the dynamic prediction for two distinct inputs coming from different records.

CHRONOS handles static and short-term dynamic aspects in separate spaces. Following the same logic, this new temporal dimension should be projected utilizing a distinct projector in a different temporal context and computing the long-term loss function there. However, the proposed methodology incorporates this loss function between the two previously mentioned temporal spaces, as represented in Figure 1. This is motivated by the following considerations:

1. As mentioned earlier, we regard these long-term dynamics as evolving slowly over time. It may be expected that these temporal components stay constant during a short-time record. Therefore, it is reasonable to represent them in the static space. Furthermore, several medical studies have highlighted the correlation between various risk factors and different health issues, such as smoking, body weight, and age (Attia et al., 2019a). It can be seen as evidence that each individual possesses a unique baseline health condition that dictates their susceptibility to cardiovascular diseases in the future. To better elucidate changes in short-term dynamics, it is believed that these baselines should also be present in this space.

2. Additionally, there is no inherent safeguard against both spaces reflecting the same characteristics. One might speculate that if all representations exclusively portray static aspects of the signal, the constraints imposed by the short-term loss function would still be satisfied. By introducing the cost function that encourages dissimilarity between representations in the two spaces, we effectively steer these projections to capture non-overlapping features. Effectively, by integrating the long-term loss in this way we made the projection heads to be contrasting.

This novel loss function term, as well as how it is calculated is shown in Figure 1. Both its effect and its use as a contrasting factor between the two spaces are explained in section 5.

### 3.2.2 SELECTIVE OPTIMIZATION

CHRONOS incorporates cost functions that may exhibit conflicting goals. Specifically, it aims to make the representations of two inputs both similar and dissimilar to each other simultaneously, by the $\mathcal{L}_{Static}$ and $\mathcal{L}_{LDT}$ loss functions that drive these respective objectives. This inherent contradiction in the optimization goals could potentially result in representations that hover in a middle ground, without achieving either of the two objectives. To address this challenge, CHRONOS employs a strategy, namely selective optimization, wherein it only focuses on the most relevant features of the representation when calculating each loss function. This approach ensures that each unit of the model is responsible for encoding patterns of a distinct temporal nature.

The process of determining which features of each representation to include in each loss function follows these steps: (i) It calculates the absolute difference between the values of the two representation vectors associated with each loss function, (ii) For the static loss function, it identifies and selects the features that exhibit the most coinciding values. Conversely, for both dynamic loss functions, it opts for the features that demonstrate a greater degree of inequality, and (iii) Any remaining features in each projection are masked and their gradients are not considered. This process is represented in Figure 1. The effect of this selective optimization is discussed in section 5.

To enhance the efficacy of our selective optimization approach, we strive to ensure that each feature within the representations encapsulates a distinctive and individual pattern. To achieve this objective, we introduce the Covariance Loss function ($\mathcal{L}_c$) as a regularization factor, effectively penalizing redundancy within the representations. This particular cost function, which has also found application in prior studies like Barlow Twins (Zbontar et al., 2021) and Variance-Invariance-Covariance Regularization (VICReg) (Bardes et al., 2022), operates as follows:

$$\mathcal{L}_c(\zeta) = \frac{1}{d} \sum_{i \neq j} [C(\zeta)]_{i,j}^2, \tag{5}$$

where $C(\zeta)$ is the covariance matrix computed on the student projection, $\zeta$. The effect of this loss function is studied in Section 5.

## 4 EVALUATION

The proposed model has undergone extensive simulation studies to evaluate its performance. Initially CHRONOS was assessed by comparing it to four state-of-the-art (SOTA) methods across four distinct downstream tasks, each using four different databases. Furthermore, we conducted two experiments to substantiate the core concept of our approach, which revolves around identifying static and dynamic patterns within time-series data with the aim of demonstrating CHRONOS' consistent capability to extract these patterns across various subjects and databases.

### 4.1 COMPARISON AGAINST STATE-OF-THE-ART TIME-SERIES SSL METHODS

CHRONOS' performance has been compared against the four most relevant SOTA methods, namely; (i) PCLR (Diamant et al., 2022), (ii) DEBS (Atienza et al., 2023), (iii) Mixing-Up (Wickstrøm et al., 2022), and (iv) TF-C (Zhang et al., 2022). Moreover, we have included the standard Bootstrap Your Own Latent (BYOL) Grill et al. (2020) framework as a baseline, focusing on capturing similarities. We have optimized the same model used in this work, under the same configuration (optimizer, data, batch size, and number of iterations), except for the TF-C method, where their proposed model has been used. This is due to the fact that it requires the use of two encoders instead of one. Note that the TF-C model contains approximately 32 million parameters, which is 30x more than the proposed CHRONOS model. To ensure that the model converges, the latter has been optimized over 75K iterations, instead of the 25K iterations proposed in this work.

Table 1: Static Tasks Evaluation

| SSL METHOD | GENDER IDENTIFICATION | | | AGE REGRESSION |
| --- | --- | --- | --- | --- |
| | ACCURACY (%) | MALE (%) | FEMALE (%) | MAE Error |
| Mixing-Up | 70.4 ±1.5 | 68.0 ±3.3 | 71.9 ±2.3 | 7.61 ±0.37 |
| TF-C | 65.8 ±2.9 | 62.8 ±5.5 | 67.4 ±2.1 | 8.57 ±0.37 |
| PCLR | 74.5 ±1.5 | 72.7 ±2.4 | 75.8 ±2.8 | **7.43 ± 0.22** |
| BYOL | 74.8 ±2.2 | 73.0 ±3.1 | 75.9 ±3.0 | 7.53 ±0.48 |
| DEBS | 68.2 ±1.6 | 66.0 ±2.9 | 69.3 ±2.6 | 8.31 ±0.18 |
| CHRONOS | **76.1 ± 2.9** | **74.2 ± 3.9** | **77.3 ± 3.2** | 7.61 ±0.21 |

To ascertain the model's versatility, we undertake four downstream tasks: Age Regression, Gender Classification, Atrial Fibrillation (AFib) Classification, and the Physionet Challenge 2017. For the first task, a simple linear regression model is fitted on top of the representations, while for the following three classification tasks, a Support Vector Classificatier (SVC) (Platt, 2000) of degree 3 and an RBF kernel is used. It is worth noting that static patterns will hold greater significance for the first two tasks, while the encoding of dynamic patterns will be pivotal for the latter two. Additionally, we draw from four different databases to assess the method's robustness across various data sources, i.e, MIT-BIH Arrhythmia Database (MIT-ARR) (Moody & Mark, 2001), MIT-BIH Atrial Fibrillation Database (MIT-AFIB) (Moody & Mark, 1983), Physionet Challenge 2017 (Cinc2017) and Sleep Heart Health Study (SHHS) (Quan et al., 1998). While the first three are publicly available in Physionet (Goldberger et al., 2000), the latter is publicly available in National Sleep Research Resource (NSRR) (Zhang et al., 2018).

**Gender Classification and Age Regression:** We randomly selected 1549 ECG time series strips of length 10 seconds, each associated with a distinct subject, from the SHHS database. We conducted a five-fold cross-validation for both downstream tasks, and the outcomes of these experiments are presented in Table 1. It can be seen how CHRONOS obtains the best metrics in three of the four cases while obtaining comparable results in which PCLR performs the best. (See Appendix D)

**AFib Classification:** We employed 10 10-second ECG time series strips originating from the eight subjects affected by AFib, belonging to the MIT-ARR database for training the ML model on top of the representation. We subsequently evaluated this model using the complete MIT-AFIB database. It is important to emphasize that the training and validation sets originate from distinct sources, therefore there is no subject overlap between them. This overlapping would significantly simplify AFib identification. The outcomes of this experiment are tabulated in Table 2. Although DEBS obtains the best metrics in this experiment, CHRONOS demonstrates a comparable performance.

Table 2: AFib identification evaluation. MIT ARR → MIT AFIB

| SSL METHOD | ACCURACY (%) | SENSITIVITY (%) | SPECIFICITY (%) |
|---|---|---|---|
| Mixing-Up | 65.6 | 60.6 | 67.4 |
| TF-C | 71.8 | 64.8 | 76.5 |
| PCLR | 73.2 | 65.6 | 78.9 |
| BYOL | 66.9 | 60.8 | 69.9 |
| DEBS | **77.5** | **75.6** | **79.5** |
| CHRONOS | 75.6 | 70.8 | 78.5 |

**Physionet Challenge 2017:** In this final experiment, we utilized the Cinc2017 database, which categorizes ECG time series strips as either Sinus Rhythm (SR), AFib, or Others. We used the dataset's pre-defined partitioning of "train" and "validation" sets. The findings of this experiment are summarized in Table 3, wherein CHRONOS achieves superior performance in three out of the four scenarios.

Table 3: Physionet Challenge 2017

| SSL METHOD | ACCURACY (%) | SR PRECISION(%) | AFIB PRECISION (%) | OTHER PRECISION (%) |
|---|---|---|---|---|
| Mixing-Up | 74.9 | 74.3 | 85.4 | 71.0 |
| TF-C | 60.2 | 62.0 | **100.0** | 44.3 |
| PCLR | 76.8 | 74.8 | 89.4 | 76.6 |
| BYOL | 78.0 | 74.8 | 88.2 | 83.3 |
| DEBS | 70.4 | 69.9 | 78.0 | 67.3 |
| CHRONOS | **78.8** | **75.5** | 90.1 | **83.8** |

**Discussion of the results:** This study has demonstrated that by considering the static, short-term, and long-term dynamics during the training process, we can drive the model to encode an information-dense representation. It leads to a consistent and performing model, which can be used in a wide spectrum of downstream tasks, using different types of datasets. Compared to related methods, CHRONOS is the only method that exhibits this capability.

We were surprised by the subpar performance of TF-C. We infer that the data augmentation techniques utilized during the training procedure are specifically tailored for EEG signals, the kind of data for which it is trained, and do not adapt as effectively to other signal domains. While we have not explored the option of selecting a distinct set of data augmentation operations better suited for ECG signals, since this is not the purpose of this paper, we posit that employing data augmentation produces a bottleneck in the adaptability of SSL methods across different domains. Moreover, given that the most favorable results are achieved by methods that do not utilize data augmentation, it suggests that its utilization should be avoided for time series, as claimed in the DEBS study.

## 4.2 REPRESENTATION STUDY

To assess the hypothesis that CHRONOS effectively encodes patterns from diverse natures, we designed the following experiment. From the MIT-AFIB dataset, we extract the representations of the 22 recordings with approximately 10 hours length. The values of each feature contained in the representations are normalized to ensure uniform value ranges. It can be assumed that static features should have a lower variance value, whereas the dynamics should have a higher one. Therefore, variance is used as a measure of whether the different features exhibit static or dynamic behavior. For each recording, the variance of the different values of distinct features is computed. We clustered the 33% of the features with the least variance values as the features that represent the static patterns of each recording. Similarly, 33% of the features with the highest variance for the cluster represent those that capture the dynamic nature. Finally, we tallied the occurrences of each feature in both clusters for the distinct records. It is expected that the same features of the representation will express the static nature consistently across distinct subjects. Therefore, the same features should be present in the different clusters consistently across the different records. This underlying principle also extends to dynamic components.

Additionally, a SHAP Analysis (Lundberg & Lee, 2017) is carried out for inferring the importance of the features in two downstream tasks. The aim of this is to study whether the static and short-term

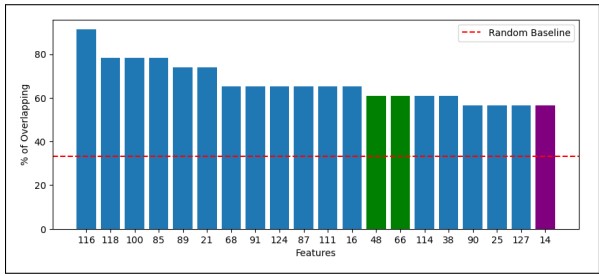 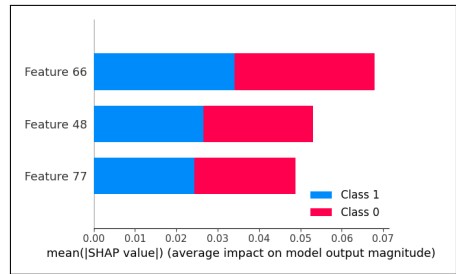

(a) Overlap of static features across distinct subjects.  (b) Features importance. Gender Classification

Figure 2: Static Features Study. The green bars represent the features that are included as the most important ones for the Gender Classification task. The purple bar represents the feature which is included within the most important ones for AFib Classification task.

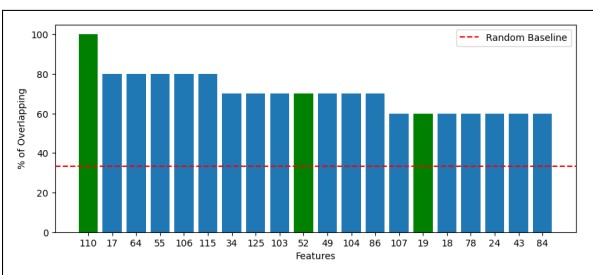 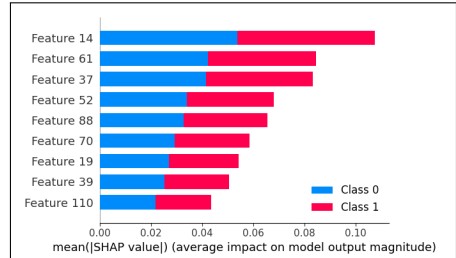

(a) Overlap of dynamic features across distinct subjects  (b) Features importance. AFib Classification

Figure 3: Dynamic Features Study. The green bars represent the features that are included as the most important ones for the AFib Classification task.

dynamic representation features play an important role. The Gender Classification task and the AFib Identification tasks are used for this purpose. It is decided to correlate short-term dynamic features with the detection of AFib, even though subjects tend to stay in both SR/AFib states for a long time. This is the reason for just selecting the 10 recordings with the most uniform distribution across the states instead of the whole database for this experiment (See Appendix E). For this SHAP Analysis, we only include the features with significant relevance, i.e., with an average impact greater than 0.04. It is important to mention that not all static features should be correlated with gender information since they can be correlated with other static aspects. But all important features for this task should be static. The same logic applies to AFib identification.

**Static Features Exploration:** Figure 2a shows the 20 features that appear most often as "static features" for each recording. The ratio of appearances goes from 91.3% to 56% so we can give it a statistical value since in a random baseline, the number of appearances would be around 33%. Figure 2b shows how 2 of the 3 most important features are included among the static features in the MIT AFIB dataset. We can consider this result of statistical relevance, since in a random baseline, only 0.48 features would be included in the 20 static ones.

**Dynamic Features Exploration:** Figure 3a shows the 20 features that appear most often as "short-term dynamic features" in each recording. The ratio of appearances goes from 100% to 60%. We studied the importance of the features using the MIT-ARR and Cinc2017 databases. Figure 3b shows how 3 of the 9 most important features are included among the so-called "short-term dynamics" in the used subset of MIT-AFIB dataset. This corresponds with more than 2 times the expected occurrences in a random baseline scenario, where only 1.4 features would be included.

**Discussion on the results:** The results obtained support the hypothesis that CHRONOS is able to consistently encode patterns of different temporal natures. It can be assured that (i) different model

units extract different temporal natures consistently across different records (Figure 2a and 3a), and (ii) the most relevant components for carrying out a static task, i.e., gender classification, exhibit a static nature across different databases. This not only validates that static patterns are consistently extracted using different databases but also that they play an important role.

Three of the most important representation components present a short-term dynamic nature, when evaluated in MIT-AFIB dataset. Finally, it can be seen that the most important feature for AFib detection exhibits a static nature (Feature 14, highlighted in Figure 2a). Although at first sight, this may seem contradictory, it only corroborates the hypothesis reflected in the Attia et al. (2019a) work, where distinct patterns are found for distinguishing the subjects that are susceptible to suffering cardiovascular disease. In addition, it supports the importance of long-term loss for enhancing these differences from the same subject over time.

## 5 ABLATION STUDY

We study the effect of (i) the incorporation of the Long-Term Loss, (ii) the use of this function as a contrasting factor between the two projections instead of creating a new projection, (iii) the Selective Optimization, and (iv) the proposed distribution of features in such optimization. This ablation study is carried out and Table 4 demonstrates how dispensing the distinct CHRONOS components mitigates the performance of the representations in the different downstream tasks.

The inclusion the $\mathcal{L}_{LTD}$ to capture long-term dynamics notably enhances model performance, particularly in Physionet 2017 Classification (Row 3 in Table 4). This supports the idea that these dynamics have been integrated into the model's representation. Moreover, employing this loss function as a contrasting term, as detailed in Section 3.2.1, not only incorporates these dynamics but also enhances other temporal aspects representations, thereby improving the quality of the representation.

Table 4: CHRONOS components Ablation study. The first row shows the proposed configuration.

| Long Term Loss | Long Term Loss as Contrasting Factor | Static/Short-Term/Long-Term Ratio (%) | | | Gender Classification Accuracy (%) | Age Regression MAE | MIT ARR ->MIT AFIB Accuracy (%) | Physionet 2017 Accuracy(%) |
|---|---|---|---|---|---|---|---|---|
| ✓ | ✓ | 25 | 50 | 25 | 76.1 | 7.61 | **75.6** | **78.8** |
| ✗ | ✗ | 50 | 50 | - | 75.1 | 7.6 | 68.8 | 72.4 |
| ✓ | ✗ | 25 | 50 | 25 | 75.3 | 7.65 | 64.7 | 77.4 |
| ✓ | ✓ | 100 | 100 | 100 | 73.3 | 7.9 | 60.3 | 75.6 |
| ✓ | ✓ | 50 | 25 | 25 | **76.5** | **7.42** | 72.4 | 74.9 |
| ✓ | ✓ | 25 | 25 | 50 | 76.3 | 7.45 | 68.7 | 71.5 |

We have also made a study of the importance of incorporating $\mathcal{L}_c(z)$ as part of the objective function. Table 5 shows how, although it does not by itself lead the model to capture useful representations, it is indispensable when it comes to leveraging the other cost functions.

Table 5: Covariance Loss Function Ablation study.

| Method | Gender Classification Accuracy (%) | Age Regression MAE | MIT ARR ->MIT AFIB Accuracy(%) | Physionet 2017 Accuracy(%) |
|---|---|---|---|---|
| $\mathcal{L}_{Static}+\mathcal{L}_{STD}+\mathcal{L}_{LTD}$ | 72.3 ±3.7 | 7.74 ±0.31 | 69.2 | 75.0 |
| $\mathcal{L}_c(z)$ | 56 ±2.7 | 8.78 ±0.46 | 40.1 | 54.6 |
| Proposed Method (Both) | **76.1 ± 2.9** | **7.61 ± 0.21** | **75.6** | **78.8** |

## 6 CONCLUSION

This paper has introduced CHRONOS, which is a novel SSL method that considers different temporal characteristics that enforce the representation to contain static, short-term dynamics, and long-term dynamics patterns presented in time-series data. We have shown that by considering these different temporal characteristics, the model is able to perform consistently in four different downstream tasks, using up to four different datasets. We posit that CHRONOS can be adapted to handle diverse types of physiological time series data with minor changes since all these kinds of data share these different temporal aspects, and no specific ECG data augmentation technique has been used.

**Limitations** While we assert the potential applicability of CHRONOS to physiological data in general, we acknowledge that our experiments have been limited to the analysis of ECG data.

ACKNOWLEDGEMENT

To come...

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

## A    THE ALGORITHM

---

**Algorithm 1:** CHRONOS

---

**Input:**

    $D$, $K$ and $N$          ▷ Set of time series, Number of iterations and Batch Size

    $\mathcal{F}(x)$ and $\mathcal{F}\prime(x)$          ▷ Student Encoder and Teacher Encoder

    $\mathcal{G}(h)_s$ and $\mathcal{G}(h)_d$          ▷ Student Static and Dynamic Projectors

    $\mathcal{G}\prime(h)_s$ and $\mathcal{G}\prime(h)_d$          ▷ Teacher Static and Dynamic Projectors

    $\mathcal{Q}(z)_s$ and $\mathcal{Q}(z)_d$          ▷ Student Static and Dynamic Predictors

    $\theta$ , and $\xi$          ▷ Student and Teacher Parameters

    $opt$ and $\tau$          ▷ Optimizer and EMA update parameter

    $\mathcal{L}_{Static}$, $\mathcal{L}_{STD}$ and $\mathcal{L}_{LTD}$          ▷ Static, Short-Term and Long-Term Loss Functions

    $\mathcal{M}_{sta}(\mathcal{Q}(z), \zeta)$ and $\mathcal{M}_{LTD}(\mathcal{Q}(z), \zeta)$,

    and $\mathcal{M}_{STD}(\mathcal{Q}(z_{t-i}), \zeta_t, \mathcal{Q}(z_{t+j}))$          ▷ Masking Procedures

    $\mathcal{L}_c(z)$ and $\alpha$          ▷ Covariance Loss and Covariance Coefficient

    $w_{size}$ and $\alpha$          ▷ Windows Size and Dissimilarity Coefficient

1   **for** $k \leftarrow 0$ **to** $K$ **do**

2      $\mathcal{B} \leftarrow \{X_1, X_2^{t-i}, X_2^t, X_2^{t+j} \in D\}_{n=0}^N$          ▷ Sample $X_1, X_2^{t-i}, X_2^t, X_2^{t+j}$ from dataset

3      $i + j \leq w_{size}$

4      **for** $X_1, X_2^{t-i}, X_2^t, X_2^{t+j} \in \mathcal{B}$ **do**

5          $h_1, h_2^{t-i}, h_2^t, h_2^{t+j} \leftarrow \mathcal{F}_(X_n^{t-i}, X_n^t, X_n^{t+j})$          ▷ Student Encoder Representations

6          $\eta_1, \eta_2^{t-i}, \eta_2^t, \eta_2^{t+j} \leftarrow \mathcal{F}\prime(X_n^{t-i}, X_n^t, X_n^{t+j})$          ▷ Teacher Encoder Representations

7          $z_1^s, z_t^s \leftarrow \mathcal{G}_s(h_1, h_2^t)$          ▷ Student Static Projections

8          $z_1^d, z_{t-i}^d, z_t^d, z_{t+j}^d \leftarrow \mathcal{G}_d(h_1, h_2^{t-i}, h_2^t, h_2^{t+j})$          ▷ Student Dynamic Projections

9          $\zeta_1^s, \zeta_t^s \leftarrow \mathcal{G}\prime_s(\eta_1, \eta_2^t)$          ▷ Teacher Static Projections

10          $\zeta_1^d, \zeta_{t-i}^d, \zeta_t^d, \zeta_{t+j}^d \leftarrow \mathcal{G}\prime_d(\eta_1, \eta_2^{t-i}, \eta_2^t, \eta_2^{t+j})$          ▷ Teacher Dynamic Projections

11          $m_{sta} \leftarrow \mathcal{M}_{STA}(\mathcal{Q}_s(z_1^s), \zeta_2^s)$          ▷ Static Mask from Static Projections

12          $m_{ltd} \leftarrow \mathcal{M}_{LTD}(\mathcal{Q}_s(z_1^s), \zeta_2^d)$          ▷ LTD Mask from Static-Dynamic Projection Pairs

13          $m_{std} \leftarrow \mathcal{M}_{STD}(\mathcal{Q}(z_{t-i}), \zeta_t, \mathcal{Q}(z_{t+j}))$          ▷ STD Mask from Dynamic Projections

14          $\mathbf{l}_n^{Static} \leftarrow 0.5 \cdot (\mathcal{L}_{Static}(m_{sta} \cdot \mathcal{Q}_s(\mathbf{z}_1^s), m_{sta} \cdot \zeta_2^s) +$

15          $\mathcal{L}_{Static}(m_{sta} \cdot \mathcal{Q}_s(\mathbf{z}_2^t), m_{sta} \cdot \zeta_1^s))$          ▷ Static Loss

16          $\mathbf{l}_n^{LTD} \leftarrow 0.5 \cdot (\mathcal{L}_{LTD}(m_{ltd} \cdot \mathcal{Q}_s(\mathbf{z}_1^s), m_{ltd} \cdot \zeta_t^d) +$

17          $\mathcal{L}_{LTD}(m_{ltd} \cdot \mathcal{Q}_d(\mathbf{z}_t^d), m_{ltd} \cdot \zeta_1^s))$          ▷ Long Term Loss

18          $\mathbf{l}_n^{STD} \leftarrow (\mathcal{L}_{STD}(m_{std} \cdot \mathcal{Q}_d(\mathbf{z}_{t-i}^d), m_{std} \cdot \zeta_t^d, m_{std} \cdot \mathcal{Q}_d(\mathbf{z}_{t+i}^d)) +$

19          $\mathcal{L}_{STD}(m_{std} \cdot \zeta_{t-i}^d, m_{std} \cdot \mathcal{Q}_d(\mathbf{z}_t^d), m_{std} \cdot \zeta_{t+j}^d)) \cdot 0.5$          ▷ Short Term Loss

20          $\mathbf{l}_n^{Cov} \leftarrow \alpha \cdot \frac{1}{6} \cdot (\mathcal{L}_c(z_1^s) + \mathcal{L}_c(z_t^s) +$

21          $\mathcal{L}_c(z_1^d) + \mathcal{L}_c(z_{t-i}^s) + \mathcal{L}_c(z_t^s) + \mathcal{L}_c(z_{t+j}^s))$          ▷ Covariance Term Loss

22      **end**

23      $\partial\theta \leftarrow \sum_{n=0}^N (\partial_\theta \mathbf{l}_n^{Static} + \partial_\theta \mathbf{l}_n^{LTD} + \partial_\theta \mathbf{l}_n^{STD} + \partial_\theta \mathbf{l}_n^{Cov})$          ▷ Compute loss gradients for $\theta$

24      $\theta \leftarrow opt(\theta, \partial_\theta)$          ▷ Update Student Parameters

25      $\xi \leftarrow \tau \cdot \xi + (1 - \tau) \cdot \theta$          ▷ Update Teacher Parameters

26   **end**

---

## B  DATA PREPROCESSING

To ensure complete reproducibility of this work, this section presents a detailed description of the preprocessing steps employed for the training and evaluation databases utilized in the proposed method.

### B.1  SLEEP HEART HEALTH STUDY (SHHS) DATA SELECTION

Only the subjects which appear in both recording cycles are used during the training procedure. This leads to 2643 subjects. ECG signals are extracted from the Polysomnography (PSG) recordings. The quality of every 10 seconds-data strips has been evaluated with the algorithm proposed by Zhao and Zhang Zhao & Zhang (2018).

We use SHHS since it contains two records belonging to the same subject, and these records are delayed with a sufficient temporal gap (around 4 years) to let these long-term dynamics changes to appear. This makes this specific database special, and this is the reason that it has been the only database used during the optimization.

### B.2  DATA CLEANING

In addition, all signals from the utilized datasets were resampled to a frequency of 100Hz. Then, a $5^{th}$ order butterworth high-pass filter with a cutoff frequency of 0.5Hz was applied to eliminate any DC-offset and baseline wander. Finally, each dataset underwent normalization to achieve unit variance, ensuring that the signal samples belong to a $\mathcal{N}(0,1)$ distribution. This normalization process aimed to mitigate variations in device amplifications that may have occurred during the data collection.

## C  IMPLEMENTATION DETAILS

We use an adaptation of the Vision Transformer (ViT) Dosovitskiy et al. (2021) model for processing physiological signals. The input data is a time series of 1000 samples, which correspond to 10 seconds-length signal sampled at 100Hz. This input is split into segments of a length of 20 samples. The model counts with 6 regular transformer blocks with 4 heads each. The model dimension is set to 128, for a total of 1,192,616 trainable parameters.

The projectors and predictors in our approach are implemented as a two-layer Multilayer Perceptron (MLP). These layers have a dimensionality of 512 and 128, respectively. Batch normalization and rectified linear unit (ReLU) operations are incorporated between the two layers of each structure. The EMA updating factor ($\tau$) is set to 0.995. The window size is set to 2 minutes. We weigh the covariance loss with a factor of 0.1. We optimize the 25%, 25%, and 50% of the features for the Static Loss, Long-Term Dynamics Loss, and Short-Term Dynamics Loss respectively, during the selective optimization. We perceive short-term patterns as more abundant since they can be associated with a variety of situations, encompassing shifts in physical activity, alterations in mental states, different sleep phases, and beyond. The effect of these different ratios is discussed in section 5.

The model is trained with 10 second-length signals belonging to the SHHS dataset Zhang et al. (2018); Quan et al. (1998). (See Appendix B). The training procedure consists of 25,000 iterations. We use a batch size of 256, and Adam Kingma & Ba (2017) with a learning rate of $3e-4$ and a weight decay of $1.5e-6$ as the optimizer. The training procedure and the evaluations are performed on a local computer, with a Nvidia GeForce RTX 3070 GPU.

## D  COMMENTS ABOUT AGE REGRESSION TASK

It might appear paradoxical that, although CHONOS guides the model to capture enduring transformations like age, its performance in the Age Regression task is subpar when juxtaposed with approaches like PCLR. Nonetheless, several researches, exemplified by Chang et al. (2022) Brant et al. (2023), indicates a lack of alignment between chronological age and reflected age in the ECG signals. Divergent health-related factors pertaining to the individual can contribute to this misalignment. Moreover, exploiting age discrepancies holds potential for deducing diverse cardiovascular

conditions. To comprehensively assess this task, it is advisable to subject different models to in-depth investigations akin to the studies mentioned, rather than relying solely on a straightforward Age Regression analysis.

# E    COMMENTS ABOUT USING AFIB IDENTIFICATION FOR DYNAMICS FEATURE EXPLORATION

As mentioned in the main manuscript, subjects tend to be in a normal or AFib state for a long period of time. We can imagine other states, such as different physical activities or changes in mental states, taking place with a higher frequency. Although the study of the dynamic features would have been optimal in these tasks, the absence of labeled data prompted the use of labeled AFIb datasets. Due to this, a selection of the records with the most state changes has been made. It is assumed that the patterns capable of explaining the differences between these states will have more variance in these selected records, so they will appear in the "dynamic features" cluster. Figure 4a shows an example of a record that has been used while Figure 4b shows an example of a record that has not been used.

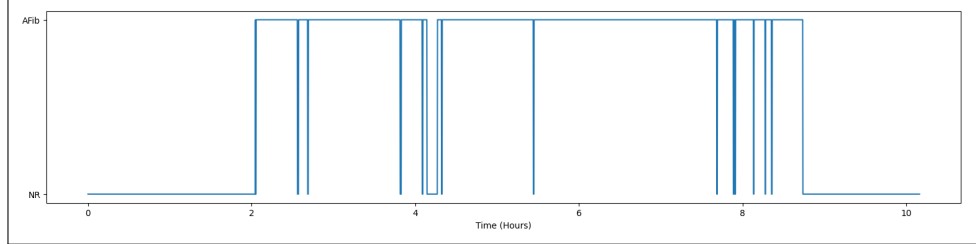

(a) Example of used record

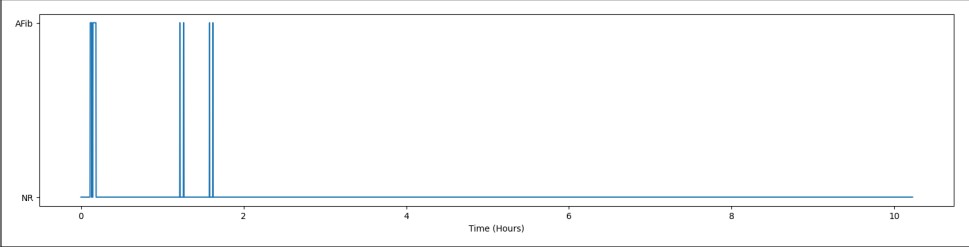

(b) Example of non-used Record

Figure 4: MIT-AFIB record selection

