# OpenReview forum: "Capturing Static, Short-Term, and Long-Term Dynamics Through Self-Supervised Time Series Learning: CHRONOS"
_ICLR.cc/2024/Conference — Submitted to ICLR 2024_

### Official Review · Reviewer_DudA · 2023-10-30

**Soundness:** 3 good
**Presentation:** 1 poor
**Contribution:** 2 fair
**Rating:** 3
**Confidence:** 3

**Summary:**

In this paper, the authors propose a new self-supervised method for medical time series. Their method, CHRONOS, aims to separate features with different temporal granularity explicitly. For that, they propose three objectives, static, short-term, and long-term dependency, using different projection heads between static and dynamic features. The authors also come up with a specific regularization scheme at training called "selective optimization" relying on masking the lowest quantile of features (or highest depending on the objective) in terms of similarity with the anchor when computing the objective. They evaluate their method on ECG data on tasks specifically relying on static or dynamic features.

**Strengths:**

### Overall useful, positive and well detailed experiments

- The experiment section (except for the feature analysis) is clear and well-organized.
- The overall performance gain with respect to the considered baseline is noticeable.
- The authors proposed an ablation study with respect to their long-term term and the usage of selective optimization
- The evaluation of tasks requiring different types of features is a great idea.

### Some interesting ideas in the method

- I appreciate the idea of using different "predictors" for different temporal granularity to better handle adversarial objectives.

**Weaknesses:**

Unfortunately, this work presents some major weaknesses which lead me to recommend for rejection.

###  Grossly insufficient related work
The literature review effort of the field from the authors is extremely small.
Between their introduction and their related work, the authors **only cited four works**. However, the field of SSL for time series is much richer. To cite a few:
- Franchesci et al. Unsupervised scalable representation learning for multivariate time series NeurIPS, 2019
- Cheng et al. "Subject-aware contrastive learning for biosignals", 2020
- Mohsenvand  et al. "Contrastive representation learning for electroencephalogram classification", Machine Learning for Health, 2020
- Tonekaboni et al. "Unsupervised representation learning for time series with temporal neighborhood coding" ICLR 2020
- Kiyasseh et al. "Clocs: Contrastive learning of cardiac signals across space, time, and patients" ICML, 2021
- Yeche et al. "Neighborhood contrastive learning applied to online patient monitoring" ICML, 2021
- Eldede et al.  "Time-series representation learning via temporal and contextual contrasting", IJCAI, 2021
- Fan et al. "Semi-supervised time series classification by temporal relation prediction" ICASSP, 2021

### Clarity in the method

- The authors don't have a part where they introduce precise notations even though the authors use a lot of different notations. This contributes to making the reading quite complicated and leaves some parts of the method unclear to me.


- In addition, the figures don't have descriptive captions. Typically, except for the colors, no description is given for the numerous components of Figure 1. Conversely, in Figures 2 and 3, no explanation is given regarding the coloring scheme.

### Some missing experiments

- The authors use a covariance loss borrowed from VICReg but do not provide ablation for that component. Hence it's impossible to know if performance improvement comes from the disentanglement of the dynamic and static features proposed by the authors or this additional regularization.
- Similarly, the authors propose to use two projection spaces instead of one because of the adversarial nature of their objectives. Unfortunately, they do not provide an experiment justifying this choice over a unique space.
- The authors introduce a window parameter for their short-term loss. They don't provide ablation for it.

**Questions:**

I have the following questions:

- Why report specificity and sensitivity and not AUROC directly? Using AUROC prevents the performance from being biased by potential miscalibration of a model.
- What is the meaning of the colored bars in Figures 2 and 3?
- Why carry constants in your loss terms that will have no impact on gradient?  "1 +  " or  "1 -"

---

> ### Author Response · Authors · 2023-11-17
> **Oficial Response to Reviewer DudA.**
>
> First of all, we would like to thank you for taking the time to read the manuscript and write the revisions.
>
> We would like to clarify the following with regard to the weaknesses mentioned above:
>
> > Regarding the grossly insufficient related work
>
> The Related Work section has been updated. The new version of the manuscript contains not only a broader view of related work, but also points out what challenges CHRONOS faces and makes it stand out from the rest.
>
> > Regarding the Clarity of the Method.
>
> In the new version of the manuscript, the first paragraph in page 3, the Figure 1 should give a clearer view of the proposed method, in addition to the notations added bellow each equation should be enough for going throgh the paper.
>
> An extended version of the captions for each figure has been incorporated to the final version of the manuscript.
>
> > Regarding the missing experiments
>
> 1. The required ablation study as been included in the Section 5.
> 2. We inherit the proposed configuration from DEBS study.
> 3. We inherit the proposed configuration from DEBS study.
>
> >  Why report specificity and sensitivity and not AUROC directly? Using AUROC prevents the performance from being biased by potential miscalibration of a model.
>
> At the time of evaluation we consider that the accuracy, specificity and sensitivity together demonstrate sufficient statistical value to compare different methods.
>
> > What is the meaning of the colored bars in Figures 2 and 3?
>
> They represent the features that are included in the set of the most important features for the respective tasks. It is reflected now in the captions of each figure.
>
> > Why carry constants in your loss terms that will have no impact on gradient? "1 + " or "1 -"
>
> The different loss terms consist of the Cosine Similarity between pairs of representations. This metric has values within a range of (-1, 1). These constants (1 + or 1 -) are added for keeping the loss values in a strictly positive range. It is a common practise when the Cosine Similarity is used.
>
> **We would like to thank you in advance for taking the time to read this rebuttal. We hope that the explanations given are sufficient to change your opinion of the work presented.**

---

> > ### Comment · Reviewer_DudA · 2023-11-21
> > **Post-rebuttal answer to authors**
> >
> > I would like to thank the authors for taking the time to work on my concerns.
> >
> > After carefully reviewing the changes made to the manuscript, I would like to maintain my score for the following reasons.
> >
> > 1) If the related is slightly richer, it still fails to provide an accurate picture of the fields of representation learning for time series and contrastive learning for medical time series.
> > 2) If the authors extended the captions, I still find the paper hard to follow and I believe it would benefit from stronger refactoring.
> > 3) The fact that the authors justify certain component choices as borrowed from a previous manuscript (DEBS) is fine to me. However, it logically diminishes their contribution with respect to this original work.
> > 4) Finally, I thank the authors for running the ablation regarding the covariance loss however, as I suspected it highlights that the performance gain compared to existing methods is not coming from the disentanglement of the dynamic and static features, weakening even more their contribution.

---

### Official Review · Reviewer_PCQf · 2023-11-01

**Soundness:** 2 fair
**Presentation:** 2 fair
**Contribution:** 2 fair
**Rating:** 3
**Confidence:** 4

**Summary:**

This paper presents an approach for feature learning from ECG signals. Specially, the author proposed to decompose the signal into three separate components: static, long-term, and short-term via a formulated contrastive loss.  Four datasets obtained from varying sources were used in their evaluation.

**Strengths:**

+ Efforts made to study learned features from their dynamics across recordings and importance in predicting for different types of learning tasks.
+ Ablation study was performed for some of the components in the proposed model

**Weaknesses:**

- The writing of the paper is very confusing. The authors described two distinct spaces, static and dynamic. However, Eq. (5) involves the calculation of the inner product between vectors from the two spaces, which indicates there is in fact only one space. Also, based on the description, it seems that the authors attempted to differentiate three types of temporal dynamics: static, short-term, and long-term. However, the proposed model only learns two types of representations, static and short-term.

- The description of the proposed method lacks clarity. For example, what is the relationship between projector and predictor and that between teacher and student networks. The proposed architecture includes encoder(s), projectors, and predictor. What exactly the representation used in downstream prediction task is not clear.

- The loss functions are not clearly motivated, or least need more explanation. For example, why two different records are used and why projection and prediction are compared while not both projections in Eq. (2). What are i and j in Eq. (3)? Hyperparameters? How were they determined?

- It seems to me that the proposed set of loss functions is problematic, leading to representation that has no discriminative capability across recordings. Eq (2) and Eq (3) drive similarity only and Eq (5) only encourages difference in learned dynamic and static features. What really helping here may be the covariance loss function (Eq. (6)), which needs more explanation and deserves an ablation study.

- The empirical results are relatively weak across the board. Statistic tests are needed in Table 1 (small difference in mean but with large variance) to show whether the differences have statistic significance.

**Questions:**

Refer to the list of weaknesses

---

> ### Author Response · Authors · 2023-11-17
> **Oficial Response to Reviewer PCQf.**
>
> First of all, we would like to thank you for taking the time to read the manuscript and write the revisions.
>
> We would like to clarify the following with regard to the weaknesses mentioned above:
>
> >  The writing of the paper is very confusing. The authors described two distinct spaces...
>
> 1. The name of the Dynamic space has been changed to short-term dynamic space in the new version of the manuscript. It should give more insights about the method.
> 2. The inner product is not calculated between the projection in the two spaces but using the predictor of the student network instead. 	This predictor can be seen as a bridge between the two spaces.
> 3. We do not know what you mean by only learning two types of representations.
> - 3a  Although it has no projection of its own, Long-Term Dynamic Loss leads the model to represent such dynamics. This is calculated between the representation of two different inputs from two different records which are delayed one from the other with a sufficient temporal gap. This loss function is motivated precisely to drive the model to capture the Long Term Dynamics.
> - 3b If you mention it for the experiments developed in section 4.2, where we have not included any that directly concerns the long-term dynamics, we believe that the mere presence of the most important feature for AFib Classification, as part of the set of static features (Section 4.2) as long as the results in the comparison against the other SOTA methods is enough to assess that we are capturing the three natures, as detailed in the discussion of results on the same section.
>
> > The description of the proposed method lacks clarity. For example, what is the relationship between projector and predictor and that between teacher and student networks. The proposed architecture includes encoder(s), projectors, and predictor. What exactly the representation used in downstream prediction task is not clear.
>
> When the original manuscript was written, it was thought that referring to the fact that the framework presented in BYOL is used as a base was enough to understand these issues. However, due to this feedback a new paragraph has been added, (last paragraph in section 3.1.1) in which it is detailed that both teacher network and the teacher projectors are updated, using EMA from their respective student pairs. It has also been added that, as is common in SSL methods, the projector and predictor are discarded after the training process. The encoder output is used during the downstream tasks.
>
> > The loss functions are not clearly motivated, ...are compared while not both projections in Eq. (2). What are i and j in Eq. (3)? Hyperparameters? How were they determined?
>
> 1. A more detailed description of the motivation of the different loss functions has been provided in the first paragraph of the page 3. This paragraph, in addition with the Figure 1, should provide more insights about the motivation of each Loss Function. The motivation of the L_LTD, which is one of the cores of this paper, is detailed in depth in Section 3.2.1
>
> 2. The student network predictions are used instead the projectors in line with the BYOL framework.
>
> 3. i and j is the time delay between the different inputs. A line has been added below the equation to reflect this. The three inputs are extracted in a two-minute window, in line with the DEBS modus operandi.
>
> >  It seems to me that the proposed set of loss functions is problematic, ... . Eq (2) and Eq (3) drive similarity only and Eq (5) only encourages .... What really helping ....
>
> 1. The required Ablation Study has been incorporated to the new version of the manuscript (Section 5). The covariance loss function by itself does not drives the model to learn useful representations.
>
> 2. As mentioned in Section 3.2.1, the Eq(5) (4 in the new version of the manuscripts), serves both for encouraging the two spaces for capturing different features, and at the same time, capturing the long-term dynamics, since it is calculated using representations from different recordings (As Illustrated in Figure 1). It enhances also differences between different recordings.
>
> 3. A new paragraph has been added in the Ablation Study explaining that the incorporation of this loss metric between different records improve the performance on the downstream tasks. Additionaly, using it also as a Contrasting Factor between the projectors improve the performance of the whole system.
>
> > The empirical results are relatively weak...
>
> There was an error in the STD given in the original manuscript for the MAE errors of the Age Regression task. They were 10x larger. We are very grateful that you brought this to our attention and it has been corrected in the original manuscript. We believe that in this way the results have statistical value.
>
>
> **We would like to thank you in advance for taking the time to read this rebuttal. We hope that the explanations given are sufficient to change your opinion of the work presented.**

---

> ### Comment · Reviewer_PCQf · 2023-12-04
> **Post rebuttal**
>
> Thanks to the authors for responding to my comments. There is still lack of convincing evidence supporting the effectiveness of the method. The empirical results are weak comparing to baseline approaches. The added ablation study does indicate covariance loss function has major impact. In addition to the lack of clarity in the writing, I keep my initial rating.

---

### Official Review · Reviewer_pVMv · 2023-11-03

**Soundness:** 2 fair
**Presentation:** 1 poor
**Contribution:** 3 good
**Rating:** 5
**Confidence:** 4

**Summary:**

The manuscript introduces the CHRONOS framework, which integrates self-supervised learning with the teacher-student paradigm for knowledge transfer, incorporating insights from novel pretraining approaches such as BYOL. The goal of CHRONOS is to extract three types of temporal information, static, short-term, and long-term, to analyze ECG signals. The authors tested their method on four tasks, including gender identification, age regression, AFib Classification, and the Physionet Challenge. The authors report a comparable performance and, in some cases, outperform the methods of Mixing-Up, TF-C, PCLR, BYOL, and DEBS.

**Strengths:**

The manuscript provides genuine ideas that have the potential to contribute significantly to the field. Concerning originality, the CHRONOS framework innovatively integrates self-supervised learning with the teacher-student paradigm for knowledge transfer, incorporating insights from novel pretraining approaches such as BYOL. Modeling, static, short-term, and long-term dynamics are not fully explored in the literature, and it might be interesting to explore the framework for other application domains, like video analysis, where temporal information is crucial.

Overall, the quality of the work is adequate, with details in the presentations, guidance to the reader, and a definition of the experimental design. CHRONOS was assessed across four independent datasets and an array of downstream tasks, showing versatility and robustness. On the other hand, despite some improvements discussed in the weakness section, the manuscripts communicate complex ideas and provide a framework that knowledgeable readers of the field can understand.

**Weaknesses:**

While the manuscript brings some exciting concepts, considerable deficiencies impact its overall quality and academic contribution. These concerns range from fundamental issues in writing and clarity to more profound matters regarding structure, content authenticity, and methodology. Below, I detail these weaknesses to provide clear guidance for necessary improvements and to ensure the work aligns with the high standards of the conference.

The manuscripts present some details related to structure and content, including:
•	The introduction lacks citations, crucial for situating the research within the existing body of knowledge.
•	The related work section needs to include more details of the current state of the field, failing to discuss current challenges and the advantages and disadvantages of existing methods. Its brevity and lack of depth need to adequately show how this new approach contributes to or differs from the state-of-the-art.
•	The absence of intuitive explanations for the proposed methodology's expected effectiveness is a significant omission.
•	The caption of the figures could be improved to be self-contained.

**Questions:**

The manuscript must be revised since minor typos and grammar mistakes impact its presentation. Also, the structure of some sentences and fluency make them difficult to follow. For example:

•	In the abstract, the authors use the word "electrocardiagram"; it should be "electrocardiogram."
•	The phrase "throughout the entire temporal sequence" is somewhat redundant. "entire" and "throughout" imply the full length of the time series. It might be more concise to say "throughout the temporal sequence" or "throughout the entire sequence.
•	The phrase: "In this paper, we introduce the Contrasting Heads Represent Opposed Natures of Signals (CHRONOS), a novel..." can be slightly improved for clarity and flow to "In this paper, we introduce the 'Contrasting Heads Represent Opposed Natures of Signals' method, hereafter referred to as CHRONOS." or "In this paper, we introduce CHRONOS (Contrasting Heads Represent Opposed Natures of Signals), a novel..."
•	The authors define the SSL and CHRONOS acronyms in the abstract, but it is considered good practice to redefine them upon their first use in the main text of the document.
•	"a SSL methodology" should be "an an SSL methodology". Remember that the choice depends on the sound that immediately follows.
•	The hypothesis is wordy and can be simplified for clarity.
•	In the introduction, the authors state "timeseries data", while it should be "time series data".
•	The phrase "...patterns belonging to three distinct temporal dynamics; static, ..." the ";" should ":".
•	Some statements are vague and biased. For example, "achieve excellent performance" and "attain good results" do not clearly understand what excellent and good means in this context and reflect the authors' opinion.

Throughout the document, several sentences need refining to improve the manuscript's clarity. While it is impractical to address each one through this medium, a comprehensive review of the entire document is recommended to improve readability and precision.

Furthermore, it would greatly benefit the manuscript to include a clear description of sections in the introduction, providing readers with a roadmap of what to expect in each area and linking it to each subsection. A short introduction in each section, especially Sections III and IV, is welcome to clarify the reader's expectations.



The manuscript must be revised since minor typos and grammar mistakes impact its presentation. Also, the structure of some sentences and fluency make them difficult to follow. For example:

•	In the abstract, the authors use the word "electrocardiagram"; it should be "electrocardiogram."
•	The phrase "throughout the entire temporal sequence" is somewhat redundant. "entire" and "throughout" imply the full length of the time series. It might be more concise to say "throughout the temporal sequence" or "throughout the entire sequence.
•	The phrase: "In this paper, we introduce the Contrasting Heads Represent Opposed Natures of Signals (CHRONOS), a novel..." can be slightly improved for clarity and flow to "In this paper, we introduce the 'Contrasting Heads Represent Opposed Natures of Signals' method, hereafter referred to as CHRONOS." or "In this paper, we introduce CHRONOS (Contrasting Heads Represent Opposed Natures of Signals), a novel..."
•	The authors define the SSL and CHRONOS acronyms in the abstract, but it is considered good practice to redefine them upon their first use in the main text of the document.
•	"a SSL methodology" should be "an an SSL methodology". Remember that the choice depends on the sound that immediately follows.
•	The hypothesis is wordy and can be simplified for clarity.
•	In the introduction, the authors state "timeseries data", while it should be "time series data".
•	The phrase "...patterns belonging to three distinct temporal dynamics; static, ..." the ";" should ":".
•	Some statements are vague and biased. For example, "achieve excellent performance" and "attain good results" do not clearly understand what excellent and good means in this context and reflect the authors' opinion.

Throughout the document, several sentences need refining to improve the manuscript's clarity. While it is impractical to address each one through this medium, a comprehensive review of the entire document is recommended to improve readability and precision.

Furthermore, it would greatly benefit the manuscript to include a clear description of sections in the introduction, providing readers with a roadmap of what to expect in each area and linking it to each subsection. A short introduction in each section, especially Sections III and IV, is welcome to clarify the reader's expectations.


The manuscript has similarities with previous studies, especially the DEBS study. Can you clarify the novel aspects of CHRONOS that distinguish it from the prior work? What specific innovations does CHRONOS introduce?

It has been noted that certain sections closely resemble sections of the DEBS paper on ArXiv without clear referencing. How do you address this similarity, and can you ensure that all reused content is properly credited?

**Details Of Ethics Concerns:**

A considerable portion of the manuscript, notably in the related work and implementation details sections, is strikingly similar to the DEBS study. While the idea and novelty might not be directly compromised, this resemblance necessitates explicit acknowledgment and proper citation to avoid academic integrity concerns.

---

> ### Author Response · Authors · 2023-11-17
> **Oficial Response to Reviewer pVMv.**
>
> First of all, we would like to thank you for taking the time to read the manuscript and write the revisions.
>
> We would like to clarify the following with regard to the weaknesses mentioned above:
>
> > The introduction lacks citations, crucial for situating the research within the existing body of knowledge.
>
> We appreciate this feedback and agree with it. In the new version of the manuscript, citations have been incorporated in the introduction. Hopefully it will help to contextualize this manuscript within the existing body of knowledge.
>
> > The related work section needs to include more details of the current state of the field, failing to discuss current challenges and the advantages and disadvantages of existing methods. Its brevity and lack of depth need to adequately show how this new approach contributes to or differs from the state-of-the-art.
>
> Again, we agree with this point. The Related Work section has been updated. The new version of the manuscript contains not only a broader view of related work, but also points out what challenges CHRONOS faces and makes it stand out from the rest.
>
> > The absence of intuitive explanations for the proposed methodology's expected effectiveness is a significant omission.
>
> After rereading the original manuscript with fresh eyes, we agree on this point. The first paragraph of page 3 has been incorporated in the new version of the manuscript for the purpose of clarifying the explanation of the method. We trust that this paragraph, in aid of Figure 1, will improve the overall understanding of the manuscript.
>
> > The caption of the figures could be improved to be self-contained.
>
> The captionsof the new version of the manuscript have been expanded to provide more information about what they represent.
>
> > The manuscript has similarities with previous studies, especially the DEBS study. Can you clarify the novel aspects of CHRONOS that distinguish it from the prior work? What specific innovations does CHRONOS introduce?
>
> Section 3 has been divided into revisiting DEBS paper, and the new features CHRONOS introduces. While it is true that this first subsection is not new to the existing body of knowledge, as it is inherited from DEBS, we believe it is of crucial importance in understanding the paper. We hope that in this new version all this will be much clearer.
>
> **We would like to thank you in advance for taking the time to read this rebuttal. We hope that the explanations given are sufficient to change your opinion of the work presented.**

---

### Official Review · Reviewer_S57d · 2023-11-06

**Soundness:** 2 fair
**Presentation:** 2 fair
**Contribution:** 2 fair
**Rating:** 3
**Confidence:** 3

**Summary:**

The paper introduces a new self-supervised learning (SSL) approach called CHRONOS for time series analysis, aimed at capturing static, short-term, and long-term dynamics in time series data. CHRONOS distinguishes itself by projecting data representations into two spaces and optimizing selective model units for each temporal aspect. The method has been evaluated on electrocardiogram (ECG) signal tasks such as arrhythmia detection, gender identification, and age estimation, showing it can surpass existing methods.

Key concepts include the identification of unique static patterns as biometric identifiers, the volatility in short-term dynamics for immediate physiological changes, and the gradual change observed in long-term dynamics related to slow-evolving characteristics like age or cholesterol level. CHRONOS extends the DEBS approach by using three types of loss functions for distinct temporal representations and captures the evolution over significant time gaps, enhancing the ability to understand and utilize physiological signal patterns for diverse health-related analyses.

**Strengths:**

1/ The authors introduce a new self-supervised learning method that incorporates observations about the nature of temporal data on medical ECG-based tasks which show three different major characteristics: static, short-term dynamics and long-term dynamics. By introducing multiple projection heads for contrasting the static and dynamic characteristics the model is able to learn more about the importance of different features for multivariate time series predictions.
2/ The authors explain the implementation and evaluation setup in detail, showing empirical performance comparisons that have an increase in performance against previous work.

**Weaknesses:**

1/ The authors frame the work as generalizing to other applications besides medical ECG however it seems most of the motivation for the design of CHRONOS is very grounded in the specifics of ECG datasets and the provided tasks with references to medical-specific features, model selection, and dataset characteristics. An expansion of the evaluation to standard baselines like UCR, or the FD-a and FD-b tasks used to evaluate TF-C, could help show the proposed generalization characteristics of this approach with more empirical validation.
2/ There are a number of existing time-series specific transformer applications. FormerTime (Cheng, et. al., 2023), for example, also focuses on multi-scale time series dynamics for multivariate time series classification, albeit without self-supervised learning. The authors should consider evaluations against these time-series specific transformer models such as Formertime and Informer to see if their proposed method continues to add benefit or if simply updating the encoder they use could lead to improvements across all the tasks evaluated using standard self-supervised learning.
3/ The figures with feature importance and overlap of features for the static and dynamic cases could benefit for my explication of which features were included (e.g. by including the feature name instead of its number).

Nits:
1/ All the references are improperly included in the text. I believe you should change from using `citet` to `citep` in most cases. References should include the author names in the sentence (e.g. “Additionally a SHAP Analysis *as proposed by* Lundberg & Lee (2017) is carried out”) or by including the author names in the citation (e.g. “Additionally a SHAP Analysis (Lundberg & Lee, 2017)...”). Please read the “Citations in Text” part of the provided LaTeX template for more information.
2/ “Short-term” vs “short term” and “long-term” vs “long term”.

**Questions:**

1/ Could the authors expand on the implications of the ablation study conducted in section 5? For example: it’s not clear why the removal of the long term loss was only evaluated when the time scale ratio was set to 25/50/25 (basides this being the ratio chosen during selective optimization). Would the long term loss add more benefit in the case where the long-term ratio is higher than 25?
2/ As an additional ablation, have the authors considered removing the selective optimization and randomly selecting a loss function at each step of the pre-training process?
3/ In the SSL time series literature, pre-training on the Sleep Heart Health Study is quite common. Why do the authors believe this dataset is more or less useful for pretraining given the tasks at hand? Have the authors tried pre-training on any other datasets?

**Details Of Ethics Concerns:**

I have some concerns about the similarity between section 3.1.1 in this paper and the equivalent section 3.1 in the DEBS (https://arxiv.org/pdf/2309.07526.pdf) paper (both titled “Non-Contrastive Method”). The paragraphs in the provided paper seem strongly inspired by the former without providing any reference to it.

---

> ### Author Response · Authors · 2023-11-17
> **Oficial Response to Reviewer S57d.**
>
> First of all, we would like to thank you for taking the time to read the manuscript and write the revisions.
>
> We would like to clarify the following with regard to the weaknesses mentioned above:
>
> > The authors frame the work as generalizing to other applications besides medical ECG however it seems most of the motivation for the design of CHRONOS is very grounded in the specifics of ECG datasets and the provided tasks with references to medical-specific features, model selection, and dataset characteristics. An expansion of the evaluation to standard baselines like UCR, or the FD-a and FD-b tasks used to evaluate TF-C, could help show the proposed generalization characteristics of this approach with more empirical validation.
>
> We believe that there has been a potential confusion in the statement that CHRONOS may be generalizable to other types of time series data, especially for other types of physiological signals. This statement is motivated by the fact that all these signals share the three dynamics detailed in this study, and no technique (i.e., data augmentation) specific to ECG signals is used. This is why we are invited to think that a model can be optimized using CHRONOS to understand other types of data, using another dataset. However, we do not assure that the model optimized using ECG signals will be able to perform on other types of signals directly. We are sorry for the confusion and sections 6 has been modified to avoid this problem of understanding.
>
> > There are a number of existing time-series specific transformer applications. FormerTime (Cheng, et. al., 2023), for example, also focuses on multi-scale time series dynamics for multivariate time series classification, albeit without self-supervised learning. The authors should consider evaluations against these time-series specific transformer models such as Formertime and Informer to see if their proposed method continues to add benefit or if simply updating the encoder they use could lead to improvements across all the tasks evaluated using standard self-supervised learning.
>
> This paper shows a new SSL method for optimizing a DL model. It means that what it is evaluated here is how SSL different methods make the same model to understand better a specific kind of data. SSL methods are evaluated on generic DL models, commonly using ResNet or ViT (as this paper). It can be supposed that the application of the presented method to a specific architecture tailored to a particular kind of data can lead to a better performance. However, the model architecture is not a topic for this paper.
>
> > The figures with feature importance and overlap of features for the static and dynamic cases could benefit for my explication of which features were included (e.g. by including the feature name instead of its number).
>
> The features you refer to compose a vector of dimension 128, which is the output of the encoder (the model being optimized). There are no names to provide further insight into what they represent. That is why their position in this representation is used during this study.
>
> > All the references are improperly included in the text. I believe you should change from using citet to citep in most cases. References should include the author names in the sentence (e.g. “Additionally a SHAP Analysis as proposed by Lundberg & Lee (2017) is carried out”) or by including the author names in the citation (e.g. “Additionally a SHAP Analysis (Lundberg & Lee, 2017)...”). Please read the “Citations in Text” part of the provided LaTeX template for more information.
>
> We agree on this point, the relevant changes have been made in the new version of the manuscript, which you can see since it has been uploaded.
>
> > About the questions you have asked, we have the following responses
>
> 1. We would like to thank you for this point, as there was an error in Table 4 in the original manuscript. Obviously when L_LTD is not taken into account, the split ratio is 50/50 over the other two. Regarding the second point and increasing the L_LTD ratio, in this same table (last row) the experiment you request has been carried out, showing that it does not improve the proposed configuration.
>
> 2. We have not considered this option, but our thinking is that it would confuse rather than help the optimization process.
>
> 3. We use SHHS since it contains two records belonging to the same subject, and these records are delayed with a sufficient temporal gap (around 4 years) to let these long-term dynamics changes to appear. This makes this specific database special, and this is the reason that it has been the only database used during the optimization. This explanation has been added to the Appendix (See Appendix B)
>
> **We would like to thank you in advance for taking the time to read this rebuttal. We hope that the explanations given are sufficient to change your opinion of the work presented.**

---

> > ### Comment · Reviewer_S57d · 2023-11-22
> >
> > Thank you for your response in both clarifying the application scope of this work, and resolving some of the questions I had about the work presented. I appreciate the time given to correcting the errors in the original manuscript and updating to use correct formatting and presentation. I believe this has improved its readability somewhat and is more in line with the official submission guidelines.
> >
> > After carefully reviewing the changes made to the manuscript, I will be maintaining my score due to the following:
> > 1. While I appreciate the clarification on what parts of the paper are original work vs included as a reference to the work on DEBS I believe this reduces the novelty and value added by this work. It would make sense to spend more time developing a method that can generalize to a broader array of applications and is not so constrained to the specific area of ECG and even only seems to work on the provided database.
> > 2. I agree with the points of reviewer DudA that the additional ablation study does not make a strong case for CHRONOS as a stand-alone method. I believe the work done in this paper can be helpful in highlighting some of the open questions in this area but requires additional development to be accepted.

---

### Author Response · Authors · 2023-11-17
**Official Comment about common complaints among reviewers.**

First of all, I would like to thank the reviewers for the feedback provided.

Due to common complaints about various points in the manuscript, it has been updated in the following points:

-	The Related Work section has been expanded. The current manuscript reflects a broader view of the state of the field. It also points out that CHRONOS is different from previous literature on the field.

-	Section 3 (Method Description) has been updated for the following purposes:

1.	It has been divided into two parts so that it is clear what is new in this work and what is a legacy of DEBS. We hope that in this way there will be no doubt of plagiarism or recycled material with this method.

2.	A new paragraph has been added, (Page 3, below Figure 1). We hope that this paragraph, together with Figure 1, will improve clarity to the explanation of the method, making it easier to understand.

-	The captions of the figures have been expanded to contain the necessary information to understand what they represent.

-	A new Ablation study has been added in which the importance of the covariance function, used as a regularization factor, is studied. Although it helps the model to learn better representations, by itself it is not able to lead the model to learn representations capable of performing in downstream tasks.


The new version of the manuscript has been upload. We hope that, guided by the revisions provided, this new version of the manuscript will conform to the quality standards required at this conference.

---

### Meta-Review · Area_Chair_SWCC · 2023-12-05

**Metareview:**

This work introduces a method for multi-head SSL on time series, focusing on learning static, fast and slow variations in signals of interest. As such, the paper is focusing on an important area, which has been studied in much detail.

The major strength of this paper is calling out the need for better SSL methods on non-stationary time series. While many papers have studied this area, there is ample room for better methodologies and benchmarking.

However, the paper has several shortcomings, most importantly a lack of situating it within the wealth of related work, various insufficiencies in the benchmarking of the method and after responding to the reviewer concerns with additional experiments, part of the benefit of the method has been reduced as well. All reviewers agree that the work is not ready to be published at ICLR in its current form.

**Justification For Why Not Higher Score:**

The paper has multiple serious flaws and is not ready for publication at ICLR, which all reviewers agree on.

**Justification For Why Not Lower Score:**

There is no lower score.

---

### Decision · Program_Chairs · 2024-01-16

Reject